# Research on Triode Based High Re-Frequency Ultrafast Electrical Pulse Generation Technology

Hantao Xu [1,2], Baiyu Liu [1,2,*], Yongsheng Gou [1,2], Jinshou Tian [1,2], Yang Yang [1], Penghui Feng [1], Xu Wang [1] and Shiduo Wei [1,2]

1   Key Laboratory of Ultrafast Photoelectric Diagnostics Technology, Xi'an Institute of Optics and Precision Mechanics, Chinese Academy of Sciences, Xi'an 710119, China; xuhantao@opt.ac.cn (H.X.)
2   Center of Materials Science and Optoelectronics Engineering, University of Chinese Academy of Sciences, Beijing 100049, China
*   Correspondence: liubaiyu@opt.ac.cn

**Abstract:** The high-repeat frequency ultrafast electrical pulse generation technology is mainly based on ultrafast switching devices combined with ultrafast circuits to generate electrical pulses with repetition frequencies of several kilohertz and a rise-time of nanoseconds or even picoseconds. This technology is the basis for several research studies and is one of the key technologies that has received wide attention from various countries. The problems to be solved are high re-frequency ultrafast high-voltage pulse generation and ultra-broadband ultrafast pulse transport and circuit stability applicability, which include circuit conduction mechanism research, pulse generation time improvement and recovery time reduction. By studying the avalanche transistor high-voltage transient conduction characteristics and reducing the loss in the carrier transport process, the influence of each parameter on the output is determined, and the key factors to enhance the circuit performance are identified. This paper designs a new high-repetition frequency ultrafast electric pulse generation (UPG) circuit using pure electronics components, which consists of combining avalanche transistor model 2N2222 with a hybrid Marx structure at the same time in the pulse circuit to add filtering, fast recovery diodes and pulse cutoff and other matching techniques to make its output more stable, which can obtain higher output frequency, faster rise-time and narrower pulse widths. It has been tested that a high re-frequency ultrafast high-voltage electrical pulse signal with a pulse repetition frequency of 200 kHz, a leading edge of 800 ps, a half-high pulse width of 5 ns, an amplitude of 1.2 kV and jitter of less than 5% can be generated at the load with a 50 Ω load at the output. The signal can be applied in the fields of ultrafast diagnosis, information countermeasures and nuclear electromagnetic radiation research.

**Keywords:** ultrafast diagnosis; high frequency; electric pulse; avalanche transistor; Marx; fast recovery diodes

## 1. Introduction

Ultrafast diagnostic techniques [1] have been widely used in basic research, photobiology, photochemistry and defense. The need for process measurements at picosecond and femtosecond time scales has driven the development of ultrafast diagnostics. To meet this need, researchers in this field have focused on developing diagnostic techniques and devices with high temporal and spatial resolution, which has become an important direction for research in this field.

At present, a complete system has been formed for the study and measurement of ultrafast phenomena represented by variable transistor streak/fragmentation camera technology [2]. In this system, the repetitive-scan streak camera is an important technology that requires high re-frequency ultrafast high-voltage pulse generation technology to provide ultrafast time-varying electrical pulses. The pulse performance is one of the key

factors affecting important metrics such as linearity, scan range and resolution of the repetitive scan camera. Therefore, the high-repeat frequency UPG technology is one of the cores of ultrafast diagnostic technology. It is also the research direction we need to focus on [3].

At the same time, high-frequency ultrafast high-voltage pulse generation technology also plays an important role in nuclear electromagnetic radiation research and information technology military confrontation. Therefore, countries all over the world are actively upgrading their electronic countermeasures and developing information-based weapons. We need to improve not only the anti-interference and stealth capabilities of our own weapons systems, but also the detection and suppression and destruction capabilities of enemy systems in order to win this "silent" war [4]. In addition, electromagnetic interference technology based on high-frequency ultrafast high-voltage pulses can also be used for anti-interference stress testing of various types of electronic products, improving the stability and accuracy of various types of equipment working in a strong electromagnetic environment and updating the concept of interference and anti-interference technology [5].

High-frequency ultrafast high-voltage pulse generation technology based on all-solid-state implementation is highly valued in all countries. In 1981, German scholar J. Jethwa [6] and others connected 11 avalanche triodes in series and obtained a pulse voltage with an amplitude of 2 kV, a rise-time of 25 ns and a frequency of 100 Hz; In 1991, the Indian scholar S. M. Oak [7] and others combined a 3-transistor series 4-stage avalanche transistor Marx circuit to obtain an output pulse with an amplitude of 3.3 kV and a rise-time of 1.5 ns; In 1994, Indian scholar V. N. Rai [8] et al. connected 15 avalanche triplets in series and obtained high-voltage pulses with an amplitude of 2 kV and a pulse rise-time of 1 ns; In 1998, Chinese scholars such as J. Liu [9] combined an 8-transistor series 5-stage avalanche transistor Marx circuit to obtain pulses with pulse amplitude up to 4 kV and full-width half maximum (FWHM) of 7 ns; In 2010, Chinese scholar Xuelin Yuan [10] combined a 20-stage avalanche triode Marx circuit to obtain electrical pulses with an amplitude of 2 kV, FWHM of 5 ns and a repetition frequency of up to 25 kHz; In 2010, J. L. Walsh [11], a British scholar, designed pulse outputs with an amplitude of 5 kV, a leading edge of less than 20 ns and a frequency of 5 kHz; In 2015, Chinese scholars J. Tan [12] et al. connected 22 avalanche triodes in series while exploiting the theory of folded reflection transmission in cables to obtain square wave-like electrical pulses with an amplitude of 2.5 kV, a leading edge of 3 ns and FWHM of 10 ns with a repetition frequency of up to 10 kHz; In the same year, Li Jiangtao [13], a Chinese scholar, designed a 16-stage compact avalanche transistor Marx circuit, where high-voltage pulses of 2.5 kV and 3 kV with FWHM of 7 ns could be obtained on 50 Ω and 75 Ω loads, respectively, at a DC voltage supply of 300 V, while the repetition frequency could reach 10 kHz; In the same year, Ding Weidong [14], a Chinese scholar, designed a compact 6-transistor series-connected 10-stage Marx circuit, which can obtain waveform-stabilized high-voltage pulses with amplitude up to 8.4 kV and pulse rise-time of about 6 ns on a 50 Ω load when providing a DC input high voltage of 1.8 kV; In 2016, American scholar C. Jiang [15] designed pulses with an amplitude of 8 kV and a rise-time of 5 ns.

Summarizing its current development status, as shown in Table 1 below.

It can be seen that under a 50 Ω load, the output amplitude of each research team can reach 5 kV or even higher at 1 Hz frequency. There are few studies on high frequency, but a few studies can reach 10 kHz under the premise of a rise-time at 100 picosecond level.

The high re-frequency pulse generation technique requires the study of unipolar unmodulated electromagnetic waves with large electric field peaks, rise-times of the order of nanoseconds or hundred picoseconds, pulse widths of the order of nanoseconds or hundred picoseconds and repetition frequencies of 100 kHz or even MHz. First of all, it is necessary to study the avalanche transistor high-voltage transient conduction characteristics, the conduction process and carrier transport process during the transistor high-voltage transient conduction so as to determine the influence of transistor parameters on the transistor conduction characteristics and then determine the key factors introducing jitter in the

transistor conduction process. This provides the theoretical basis and experimental basis for the targeted device selection and optimization measures.

The scientific problems to be solved are high re-frequency ultrafast high-voltage pulse generation, ultra-broadband ultrafast pulse transmission and circuit stability applicability, which include studying the circuit conduction mechanism, improving pulse generation time and reducing recovery time. The design of a high-frequency ultrafast high-voltage pulse generation source based on all-solid-state implementation has not only theoretical research value, but also practical application value.

**Table 1.** Current status of development in this field.

| Year | Country | Voltage Value | Rise−Time | FWHM | Repeat Frequency | Realization Path |
|------|---------|---------------|-----------|------|------------------|------------------|
| 1981 | Germany | 2 kV | 25 ns | / | 100 Hz | Serial |
| 1991 | India | 3.3 kV | 1.5 ns | / | 1 Hz | MARX |
| 1994 | India | 2 kV | 1 ns | / | 1 Hz | Serial |
| 1998 | China | 4 kV | / | 7 ns | 1 Hz | MARX |
| 2010 | China | 2 kV | 0.4 ns | 5 ns | 25 kHz | MARX |
| 2010 | UK | 5 kV | 20 ns | / | 5 kHz | / |
| 2015 | China | 2.5 kV | 3 ns | 10 ns | 10 kHz | Serial |
| 2015 | China | 3 kV | / | 7 ns | 10 kHz | MARX |
| 2015 | China | 8.4 kV | 6 ns | / | 1 Hz | MARX |
| 2016 | USA | 8 kV | 5 ns | 4.6 ns | 500 Hz | / |

## 2. Materials and Methods

The basic principle of ultrafast high-voltage pulse generation is to control the long-time accumulation and instantaneous release of electric charge [16] by first charging the energy storage components in the circuit, such as capacitors, and then using electronic components as fast switches to change the structure of the circuit. The electrical energy stored in the devices is simultaneously released in a short period of time, and a high amplitude electrical pulse signal can be obtained in a short period of time.

### 2.1. High-Voltage Switchgear Module

The electrical pulse generation process generally includes energy storage, charging and switch conduction and discharge. The conduction switch used for discharge includes gas switch, liquid switch and solid switch. The gas switch can withstand very high voltage, great current and long-service life, but has slow conduction speed (hundred nanosecond level), large trigger jitter (nanosecond level) and low gas switch-repetition frequency; the liquid switch is mainly characterized by high withstand voltage, high conduction current and high switching power, but its disadvantages are huge volume, slow conduction speed (millisecond level), large trigger jitter (millisecond level) and low switch-repetition frequency [17].

Based on the above problems, with the rapid development of semiconductor technology, more and more solid-state switches are used in various fields. Solid-state switches are devices that use their own characteristics to achieve fast on/off functions. Solid-state switching devices applied to ultrafast high-voltage pulse generation mainly include MOSFET, avalanche transistors, step recovery diodes and IGBT [18], etc. Among them, MOSFET and IGBT conduction speed can only reach the fastest nanosecond level [19]. Although the step recovery diode conduction speeds up within a hundred picoseconds, its low withstand voltage, which is only to a few tens of volts, cannot meet the cathode selective voltage requirements of the enhancer. The avalanche transistor is a fast-switching device; its conduction speed can reach a hundred picoseconds using multiple avalanche transistors

in series, as long as the addition of equalization resistors or voltage regulator diodes can also withstand high voltage. Therefore, avalanche transistors were chosen for this design.

### 2.2. Avalanche Transistor Conduction Mechanism

Avalanche triode is a semiconductor device that utilizes the avalanche effect and differs from the general triode in that its main operating area is in the avalanche region.

They are an important component of high-speed semiconductor circuits due to their fast response, high peak power, ease of cascading and high stability. Avalanche transistors have three basic types of conduction: triggered conduction, overvoltage breakdown conduction and dF/dt conduction [20].

There are two types of trigger conduction: optical and electrical triggering. Electrical triggering is widely used in ultra-high-speed circuits. The triggering method requires external stimulation and is characterized by a long turn-on time of about 200 ns. transistor overvoltage breakdown refers to the occurrence of an avalanche transistor in which the voltage between the collector and the emitter exceeds the nominal breakdown voltage. When the avalanche transistor opens, however, the circuit cannot provide sufficient current at this time, so the avalanche transistor does not fully conduct and the amplitude of the linear region of the waveform is low. dF/dt conduction occurs when the voltage between the poles of the CE avalanche transistor is not high enough to reach the breakdown voltage, but the avalanche transistor also conducts at a high rate of change.

The avalanche transistor conduction effect can be used to generate fast rise-time, high-power electrical pulses. To increase the amplitude of the output pulse, multiple avalanche transistors are usually connected in series or in a Marx configuration [21]. By applying a high reverse voltage to the collector of an NPN crystal triode, an extremely high field strength is generated in the charge region within the collector junction. When these high-energy charges collide with the lattice, new charge carriers are generated and these free charge carriers are also accelerated by the strong electric field, repeating this process over and over again, resulting in a rapid increase of current in the collector junction in the form of an "avalanche" and an avalanche effect in the transistor. When a transistor operates in the avalanche region, it has a negative collector-emitter resistance characteristic. When the reverse bias voltage of the collector junction continues to increase, the multiplication effect leads to high current and secondary breakdown, which exceeds the linear range and has a negative resistive characteristic.

### 2.3. Pulse Generation Circuit

The output pulse amplitude of a single avalanche triode is limited, so its use is somewhat restricted [22]. In order to increase the amplitude of the output pulse, several avalanche transistors in series cascade circuit or parallel circuit are often used. Cascade circuit structure is simple and conducive to quickly generating a pulse with a large slope edge. However, with the increase in the number of cascades, the required input supply voltage also increases. If the voltage is too high, the device is likely to discharge breakdown. In contrast, the parallel scheme requires a lower supply voltage. The same power supply can supply more stages, but the circuit operates with a largely distributed capacitance, where the probability of false triggering of the device increases, the stability of the output is affected and the high-voltage power supply could also lead to damage. A new structure combining the advantages of these two circuits is called the Marx-type circuit, which has a cascade structure for each stage and multiple circuits connected together in parallel during the charging state for better access to fast, high-amplitude electrical pulses.

The number of energy storage capacitors in a Marx structure circuit must be equal to the number of stages. In the case of almost simultaneous breakdown of all avalanche triodes, the number of stages has little effect on the pulse width. The falling edge of the negative pulse depends on the breakdown time, while the rising edge depends on the RC circuit [23].

Based on the all-solid-state implementation of high-frequency ultrafast high-voltage pulse generation technology, in the early route through a serial and parallel transistor implementation, the schematic diagram in Figure 1a,b shows a serial basic structure, where the collector and emitter of multiple transistors are connected first and last. A series circuit structure is simple and has a short pulse rise-time and small distributed capacitance, but requires additional equalization circuit to maintain the current balance and a very high DC voltage supply. Its pulse generation efficiency is low, while the supply voltage is too high, making it easy to produce a high-voltage discharge firing phenomenon; transistor damage may lead to the simultaneous damage of all transistors [24]. Compared to series circuits, parallel circuits can obtain higher pulse amplitudes with a lower supply voltage. The shortcomings are that the circuit has a large distributed capacitance, a long rise-time to reach the maximum amplitude and all triodes need to be triggered completely synchronously, which makes the circuit too complex and therefore less stable and synchronized.

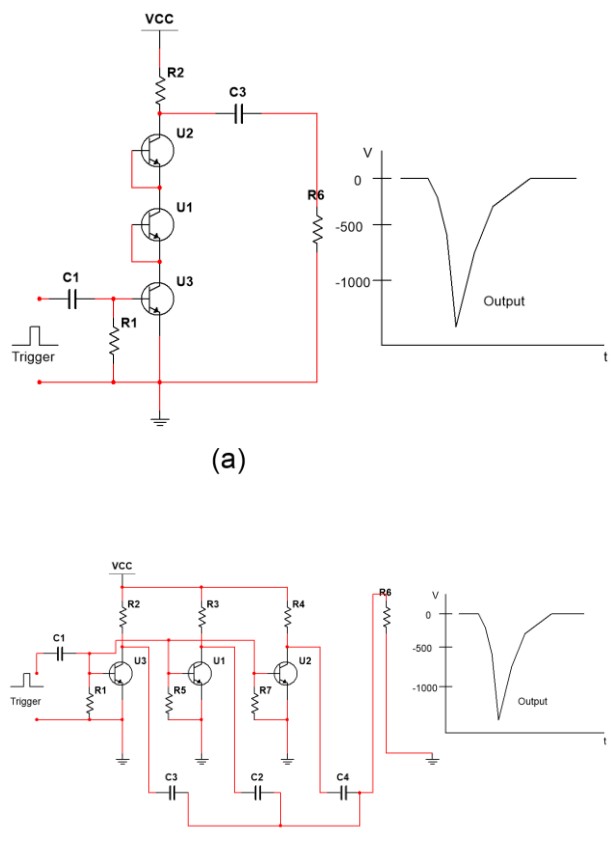

**Figure 1.** (**a**) Serial architecture and (**b**) parallel architecture.

Guided by the theoretical study and combining the advantages and disadvantages of the two circuits, this paper proposes to use a hybrid Marx cascade structure circuit to develop the high re-frequency nanosecond 10,000-volt pulse source required in this study. This circuit structure combines the advantages of serial and parallel circuits. By charging and discharging in parallel, a high pulse amplitude can be obtained at a low start-up voltage while also obtaining a small circuit parasitic parameter. This schematic is shown in Figure 2. The high-speed switch of the whole circuit consists of a large number of avalanche transistors, and the output amplitude is mainly determined by the breakdown voltage and the number of avalanche transistors.

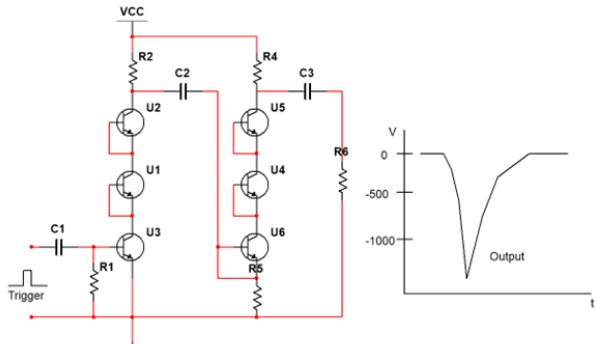

**Figure 2.** Hybrid Marx architecture.

A comparison of the three circuit structures is shown in Table 2:

**Table 2.** Comparison of different circuit structures.

| Structure Type | Advantages | Disadvantages |
|---|---|---|
| Serial Architecture | Simple structure<br>Short rise-time<br>Small distributed capacitance | High input voltage<br>Low generation efficiency<br>Easy discharge to light up<br>Transistors easily damaged at the same time |
| Parallel Architecture | Low input voltage<br>Unbreakable | Large distributed capacitance<br>Long rise-time<br>Requires synchronized triggering |
| Hybrid Marx | Low input voltage<br>Small distributed capacitance<br>High generate efficiency<br>High stability | Structural complexity<br>More parameter matching |

*2.4. Theoretical Derivation*

As shown in Figure 2, a trigger pulse is connected between the BE junctions of the first triode and the frequency of the trigger pulse can be selected as required. One cycle of the circuit operation can be divided into a charging phase and a discharging phase.

During the charging phase, the avalanche transistor is in the cutoff state, and the circuit can be equated to a circuit composed of resistance, inductance and capacitance. The power supply $V_{CC}$ charges the capacitor C until it reaches a steady state almost equal to $V_{CC}$ after Uc is no longer changing, as shown in Figure 3 below, where L is the total inductance of the discharge equivalent circuit, R is the total resistance of the discharge equivalent circuit and C is the equivalent capacitance of the discharge circuit.

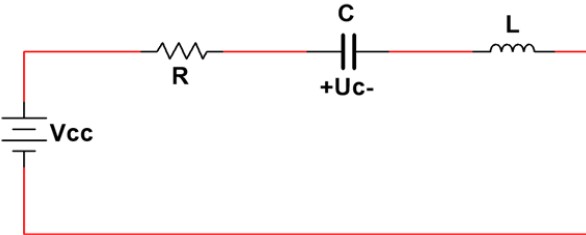

**Figure 3.** Equivalent diagram of charging circuit.

List the loop equation as [25]:

$$LC\frac{d^2u_c}{dt^2} + RC\frac{du_c}{dt} + U_C = V_{cc} \tag{1}$$

The initial value is:

$$u_C(0) = 0 \tag{2}$$

$$u_C'(0) = 0 \tag{3}$$

The first term in the judgment equation contains the total resistance *R*, taking a range of values from 100 kΩ to 10 MΩ, capacitance of pf magnitude, inductance of nh magnitude; it is estimated that the first term is much larger than the second term, that is:

$$\Delta = (RC)^2 - 4LC \gg 0 \tag{4}$$

Therefore, the general solution takes the form:

$$U_C = C_1 e^{\lambda_1 t} + C_2 e^{\lambda_2 t} + V_{cc} \tag{5}$$

Among them:

$$\lambda_1 = \frac{-RC + \sqrt{(RC)^2 - 4LC}}{2LC} \tag{6}$$

$$\lambda_2 = \frac{-RC + \sqrt{(RC)^2 - 4LC}}{2LC} \tag{7}$$

Substituting the initial value:

$$C_1 = \frac{V_{CC}}{\frac{\lambda_1}{\lambda_2} - 1} \tag{8}$$

$$C_2 = \frac{V_{CC}}{\frac{\lambda_1}{\lambda_2} - 1} \tag{9}$$

$C_1$ is negative, close to $-V_{CC}$ and less than $-V_{CC}$. $C_2$ is positive, close to 0. A substitution gives:

$$Uc = \frac{V_{CC}}{\frac{\lambda_1}{\lambda_2} - 1} e^{\lambda_1 t} + \frac{V_{CC}}{\frac{\lambda_1}{\lambda_2} - 1} e^{\lambda_2 t} + V_{CC} \tag{10}$$

Charging time is:

$$t = \frac{ln\left(-\frac{c_2}{c_1}\right)}{\lambda_1 - \lambda_2} = \frac{ln\left(-\frac{\lambda_1}{\lambda_2}\right)}{\lambda_1 - \lambda_2} \tag{11}$$

Analysis of the equation shows that the initial voltage value of the capacitor is 0. The charging phase will reach $V_{CC}$ after a time *t*. When charging, it is approximated that the triode is in a closed state, which is equivalent to a break in the circuit, and the power supply charges the capacitor whose value remains constant after stabilization. It is important to note that the capacitance *C* includes the charging capacitor in the circuit, the rectifier capacitor, the capacitor in the differential circuit and the capacitor in the device distribution parameters, meaning that the sum of the voltages obtained on all capacitors is equal to the supply voltage. In the actual circuit, we measure only the charging capacitor, which, although much larger compared to the other capacitors, will eventually stabilize at a voltage slightly lower than the supply voltage. When the charging capacitance is small, although the total equivalent capacitance is still charged to the supply voltage, the voltage across the capacitors will drop due to the difference in the storage capacity of the small capacitors, resulting in a corresponding drop in the maximum amplitude of the pulse, but the trailing edge of the pulse will be accelerated because the capacitor charge discharges faster, which is an undercharge method [26].

The discharge phase begins when the pulse trigger arrives, the avalanche transistor is in the cutoff state and the capacitor begins to charge, at which time a large number of carriers are gathered in the avalanche transistor. In the moment of being triggered by the voltage, the three-stage transistor is rapidly avalanching conduction, releasing the

charge stored in the capacitor to form a steep pulse rise-time, after reaching the peak and then after the circuit RC to form the pulse trailing edge. The equivalent diagram of the discharge loop is shown in Figure 4, where C is the equivalent total capacitance, L is the equivalent total inductance and R is the equivalent total resistance. When the pulse trigger reaches the first avalanche transistor, it will conduct, then the EB pole voltage of the second avalanche transistor will rise, making its CB two-point voltage greater than the breakdown voltage it can withstand, which results in overvoltage breakdown and conduct, and then the subsequent avalanche transistor avalanche breakdown and conduct in turn. When all the avalanche transistors are on, the original parallel capacitors become series and the accumulated charge on multiple capacitors is released in series, thus generating a fast-falling negative pulse on the load.

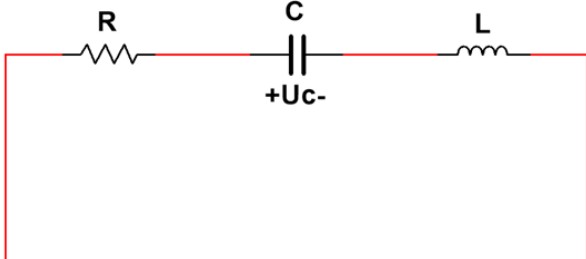

**Figure 4.** Equivalent circuit diagram of discharge circuit.

List the loop equation as:

$$LC\frac{d^2U_c}{dt^2} + RC\frac{dU_c}{dt} + U_c = V_{cc} \tag{12}$$

The initial value is:

$$u_C(0) = 0 \tag{13}$$

$$u'_C(0) = 0 \tag{14}$$

The total resistance $R$ of the first term in the judgment equation is taken to be of the order of a hundred ohms, the capacitance is of the order of pf and the inductance is of the order of nh, so we have:

$$\Delta = (RC)^2 - 4LC < 0 \tag{15}$$

The equation has a pair of conjugate complex roots and the solution takes the form:

$$U_C = e^{\alpha t}(k_1 cos(\omega t) + k_2 sin(\omega t)) \tag{16}$$

Of which:

$$\alpha = -\frac{R}{2L} \tag{17}$$

$$\omega = \sqrt{\frac{1}{LC} - \left(\frac{R}{2L}\right)^2} \tag{18}$$

Substituting the initial value:

$$k_1 = V_{cc} \tag{19}$$

$$k_2 = -\frac{\alpha V_{cc}}{\omega} \tag{20}$$

Then the voltage equation to find the current equation:

$$\frac{dU_C}{d_t} = e^{\alpha t}(D_1 cos(\omega t) + D_2 sin(\omega t)) \tag{21}$$

Of which:

$$D_1 = \alpha k_1 + \omega k_2 = 0 \tag{22}$$

$$D_2 = \alpha k_2 - \omega k_1 = -\frac{-\alpha^2 + \omega^2}{\omega} V_{cc} \tag{23}$$

So, we get:

$$i = c\frac{dU_C}{dt} = Ae^{\alpha t}\sin(\omega t) \tag{24}$$

Of which:

$$A = -\frac{V_{CC}}{\omega L} \tag{25}$$

When the trigger pulse goes low, all the avalanche transistors enter off state again and the circuit enters the charge loop state for the next charge. Depending on the trigger pulse frequency, the effective charging and discharging times of the capacitor are different, so the amplitude and width of the output pulse obtained also change. When the frequency is too high and the effective time is less than the required charging time of the capacitor, there will be a significant attenuation in the output amplitude. It should be noted that the capacitor consumes energy during charging and discharging, so the energy loss of the circuit needs to be considered in practical applications.

## 3. Results

### 3.1. Circuit Design and Implementation

When designing an avalanche transistor driver circuit, several factors need to be taken into account, including the accuracy, stability and reliability of the driver circuit. The specific design and implementation process is as follows (Figure 5).

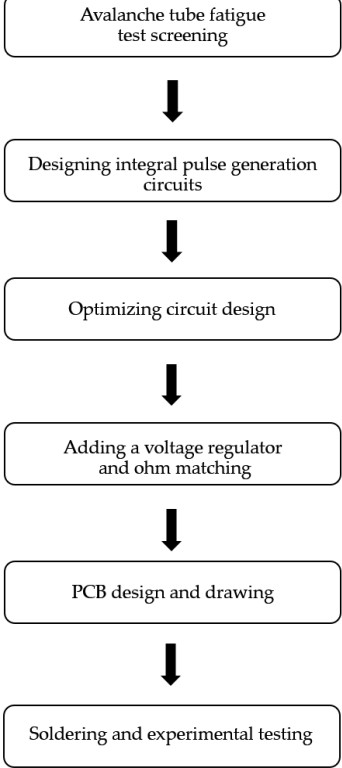

**Figure 5.** Design and implementation process.

In order to ensure the reliability of the circuit, fatigue test screening is first required, including screening and aging experiments for specific avalanche transistors in strict accordance with the parameter specifications to widen the avalanche voltage regulation

range of individual components. This results in more reliable and stable operation at the device level, improved operating consistency and reduced output jitter at the overall circuit level.

In the theoretical calculation, if the number of energy storage capacitors corresponds to the number of stages of the circuit, the capacitors are able to charge up to the supply voltage [27]. However, in the actual circuit, the voltage at final stabilization may be slightly lower than the supply voltage when the collector capacitance is relatively small, thus affecting the maximum amplitude and trailing edge speed of the pulse. Therefore, these factors need to be fully considered when designing the circuit to ensure proper operation and stability of the circuit. The basic Marx circuit schematic is shown in Figure 6.

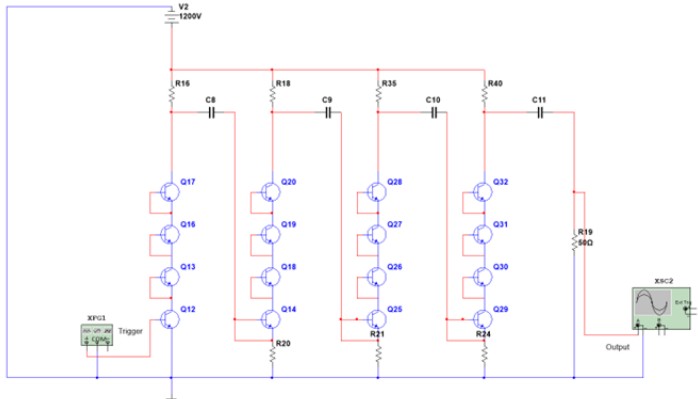

**Figure 6.** Basic Marx circuit schematic.

The improved circuit design is shown in Figure 7, with added filter capacitors at the power input and trigger input to make the power supply voltage and trigger signal more stable and to reduce burr, which can ensure a stable supply of energy and trigger at high re-frequency; adding voltage regulator resistors keep the voltage drop on each transistor balanced and increase the conduction consistency, which can speed up the rise-time to enhance the amplitude while reducing avalanche triode damage; adding EB inter-pole resistors make the BVCBO and BVCEO interval more stable, enhance the output waveform stability and effectively reduce loss and jitter at high frequencies; replacing the grounded current limiting resistors with fast recovery diode can speed up the charging speed when charging and have a better isolation protection effect when discharging, reduce energy loss and improve the re-frequency; adding a sharpening circuit at the output, as in Figure 8, using the avalanche diode's avalanche characteristics, accelerates energy release, while inductive isolation can effectively shorten the trailing edge.

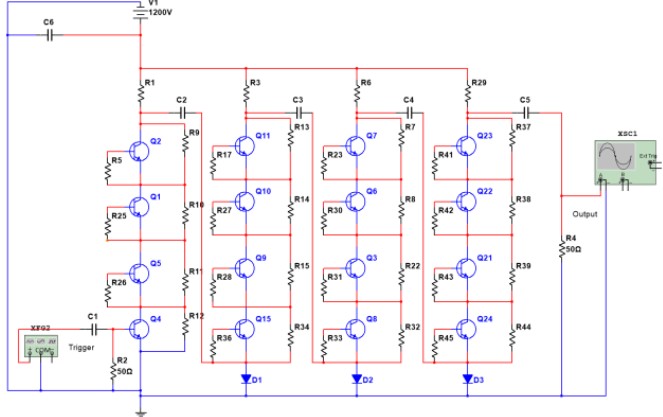

**Figure 7.** Optimized Marx circuit.

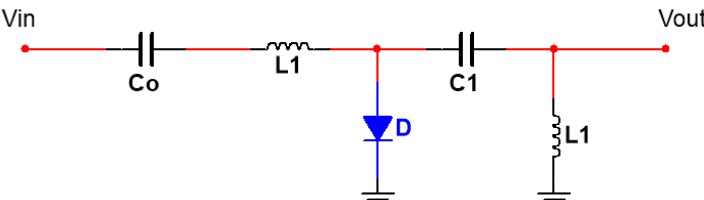

**Figure 8.** Diode sharpening circuit.

We then proceed to the circuit PCB drawing. The PCB is drawn considering stray capacitance, high-voltage isolation, overall impedance, electromagnetic interference and other factors. The filtering circuit and amplifying circuit are used to filter and amplify the interference of the trigger pulse, and the distributed circuit is designed to divide all avalanche transistors into multiple stages and ground them through fast recovery diodes to form a high-frequency electrical pulse transmission structure, which can effectively reduce the energy loss when charging and discharging the circuit, greatly shorten the rise-time and improve the recovery speed after the circuit is discharged; at the same time, the pulse output of the PCB board is designed as a Microstrip line structure, as a way to reduce high-frequency losses; finally, adding current-limiting resistors and equalization resistors to prevent the voltage from being higher than the threshold voltage and damaging the device, adding fast recovery diodes to improve the secondary charging speed after discharge, in order to make it widely used, also match the end of the whole circuit to 50 Ω. PCB physical diagram is shown in Figure 9. The design of the PCB should ensure the largest possible copper cladding, increase the uniform over-hole to reduce the impedance to ground, control the spacing to ensure the degree of electrical isolation. At the same time, choosing the appropriate pad area can effectively reduce the distribution parameters and enhance the conduction speed. As this design uses high-power resistors, heat dissipation and volume issues need to be considered.

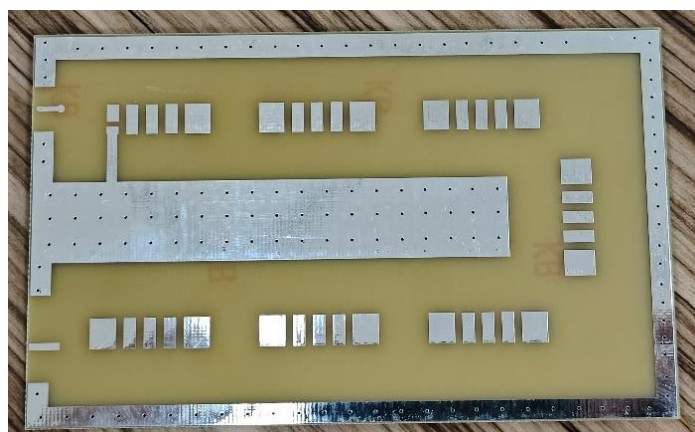

**Figure 9.** PCB physical diagram.

We selected a 900 V high-voltage DC power supply. For avalanche transistor, we selected the 2N2222 made by NXP. For current limiting resistor, we selected a 20 KΩ, For capacitor, we selected a 100 pF. For equalization resistor, we selected 10 MΩ. For fast recovery diode, we selected the FR307. We also paid attention to the current limiting resistor and fast recovery diode, the power rating and the heat dissipation of the complete board. After completing the production of the circuit board and welding work, we chose the appropriate experimental instruments for measurement. For this experiment, we used an oscilloscope, signal generator, high-power high-voltage DC power supply and 1000 times attenuator.

*3.2. Output Results*

In this experiment, we used the signal generator as the input trigger signal at an amplitude of 5 V, pulse width of 100 ns and rising edge of 10 ns pulse wave; power supply selection was of high-voltage DC power supply; to observe the waveform, pulse output went through the BNC interface connected to 50 Ω cable, accessed coaxial fixed attenuator attenuation 1000 times, and then through the 50 Ω cable to the oscilloscope.

The quality of this trigger signal is critical; its leading edge should be steep enough, the amplitude should be sufficient to trigger the avalanche transistor, the pulse width should not be too narrow, while crosstalk and interference should be reduced and the trigger pulse should be provided to the circuit using an independent device through the SMA interface. For this design, we used the AGILENT model 33250 A function/arbitrary waveform signal generator, as shown in Figure 10.

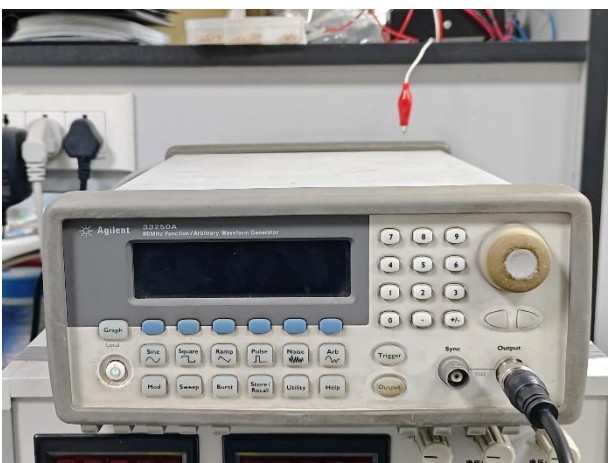

**Figure 10.** Function/arbitrary signal generator.

The power supply is a high-voltage DC bias power supply. The avalanche transistor model used in this circuit is 2N2222. The voltage value of the DC power supply is related to the number of avalanche transistors and the power supply power is related to the repetition frequency, so it needs to provide a high-power DC supply. In this design, we used a DC high-voltage power supply, model DW-P202-20F76 high-power positive high-voltage DC power supply. As shown in Figures 11 and 12, it operates from 24 V DC power supply and outputs positive DC high voltage from 0 to 2 kV.

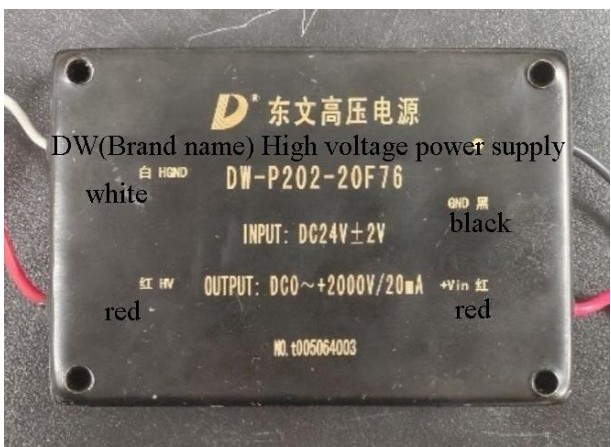

**Figure 11.** High-power high-voltage DC power supply.

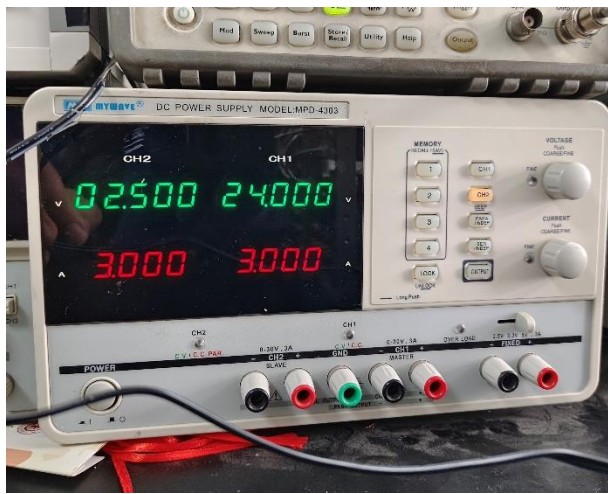

**Figure 12.** 24 V power supply.

The broadband oscilloscope used was the TEKTRONIX MS064B, as shown in Figure 13.

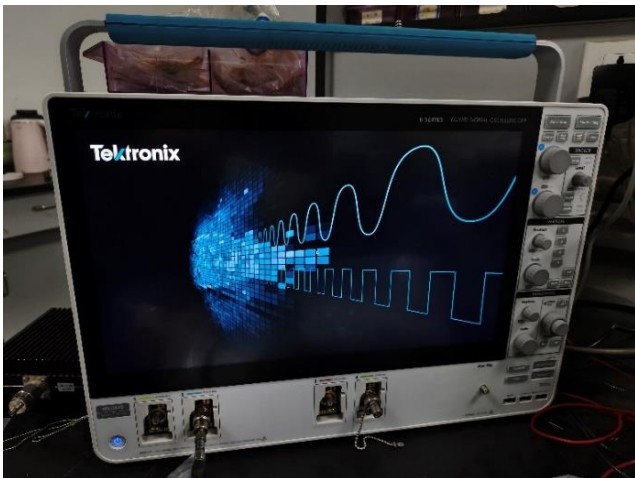

**Figure 13.** Oscilloscope.

The attenuator was HD-040CHPFA60N-100F DC-4GHz, as shown in Figure 14.

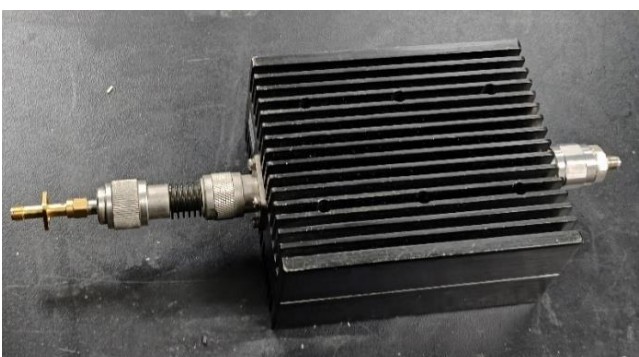

**Figure 14.** 1000 times attenuator.

The built experimental platform is shown in Figure 15.

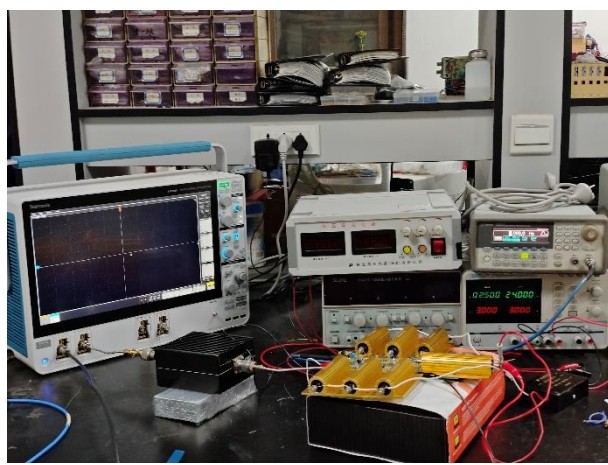

**Figure 15.** Experimental platform.

We connected the signal generator and oscilloscope and added the 1000 times attenuator between the output and the oscilloscope. We set the output of the signal generator to 1 Hz, 5 V pulse waveform, set the internal resistance of the oscilloscope to 50 Ω, set the trigger to −300 mV and observed the waveform of the single output of the pulse circuit. The experimental waveforms were exported as .csv files to a USB drive using the oscilloscope's own screenshot function, and then generated as clear, visual graphs by Origin 2023.

At the input voltage of 750 V, a high-voltage pulse of 1.2 kV was observed on a 50 Ω load, as shown in Figure 16. The oscilloscope time axis was widened to clearly observe the leading edge and FWHM of the pulse, as shown in Figure 17. It can be observed that the leading edge is about 800 ps and the FWHM is about 5 ns.

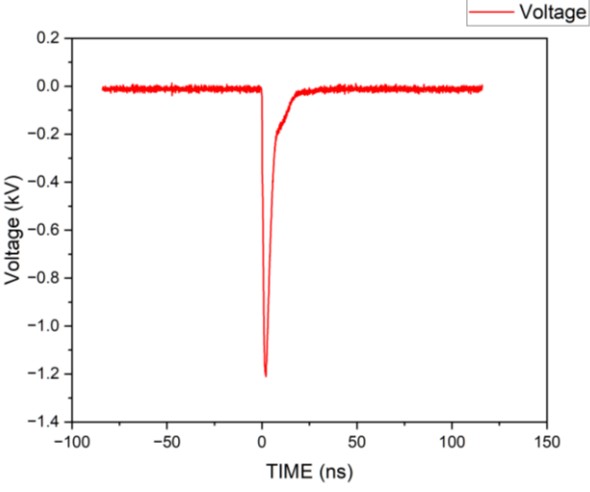

**Figure 16.** Pulse output waveform at 1 Hz (attenuation 1000 times).

Next, the output frequency of the signal generator was adjusted to verify the high-frequency capability of the circuit. The output frequency of the signal source was changed to 200 kHz and the supply voltage was adjusted to 900 V to meet the power supply requirements since higher power is required at higher re-frequencies. The time coordinate of the oscilloscope was scaled in order to observe the output period. Figure 18 shows that the pulse output interval was 5 us and the frequency was calculated to be 200 kHz to match the trigger signal. While operating at a high repetitive frequency, the individual pulse waveforms were observed again, as in Figure 19; it can be seen that there was little change from 1 Hz, and within the cumulative working time of 5 min, its jitter was less than 3%.

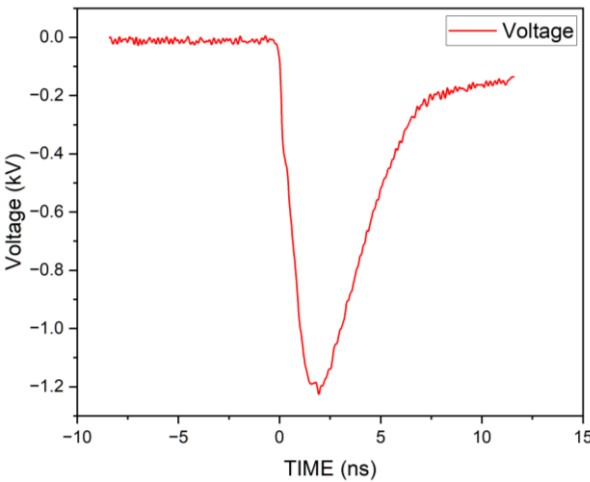

**Figure 17.** Pulse output rise-time is about 800 ps. FWHM is about 5 ns (attenuation 1000 times).

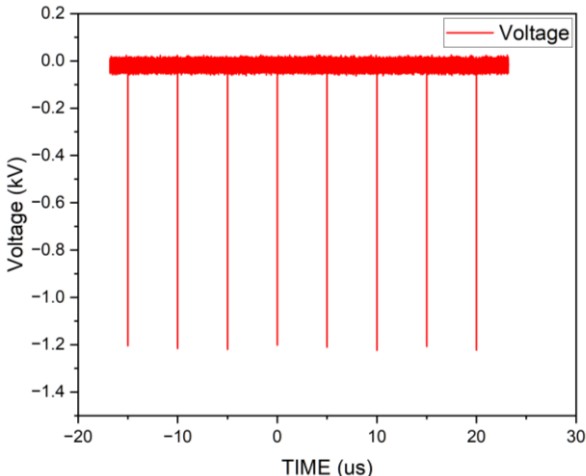

**Figure 18.** Waveform output period of 5 us, frequency of 200 kHz (attenuation 1000 times).

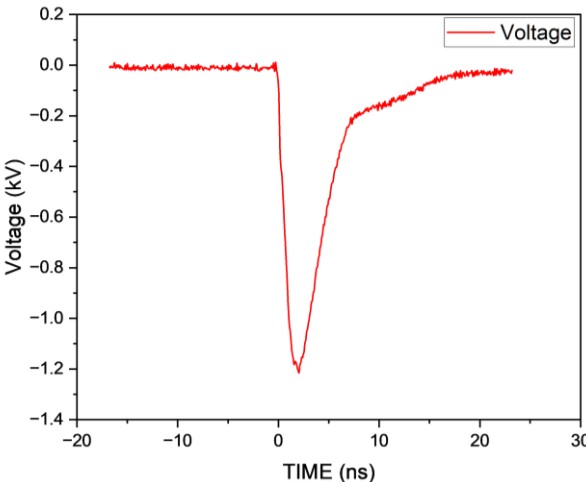

**Figure 19.** Pulse output waveform at 200 kHz (attenuation 1000 times).

In summary, this circuit can output a high re-frequency ultrafast high-voltage electrical pulse with an amplitude of 1.2 kV, a leading edge of 800 ps and FWHM of 5 ns at a repetition frequency of 200 kHz. This test result has a large improvement compared with

the traditional pulse source, breaking through a hundred thousand Hz with guaranteed amplitude and high stability.

## 4. Discussion

### 4.1. Wave-Shaped Discontinuity

From the output waveform, it can be seen that there is a large trailing. There could be two reasons for the behavior.

First, when the trigger signal arrived, the transistor went into avalanche, then the device briefly turned on and acted like a resister. After the trigger, the transistor turned off. This represents a much higher resistance from the avalanche transistor in its off state. More research must be performed in the leakage current of the device while subjected to a higher supply voltage.

Second, strayed capacitance and resistance could be arising from multiple locations in the assembled module. They can significantly affect the characteristics of the output pulse.

Based on the above observation, further investigation and subsequent optimization of the circuit should be performed.

### 4.2. High Repetition Rate

The main factors determining the maximum operating frequency of a pulse circuit are the working frequency of the avalanche transistor, the charging and discharging speed of the circuit, power supply capacity and heat dissipation conditions.

Firstly, it is the working frequency of avalanche diodes. According to Data Sheet, 2N2222 has a conduction speed of about 200 ps and an internal capacitance of 25 pF. A next step would be to select a better-performing triode. The power consumption limitation of avalanche diodes is affected by conduction current. When a single transistor approaches critical power consumption limit quickly, multiple avalanche diodes can be connected in parallel to divert their conduction current so that they operate within safe power consumption range and ensure output pulse repetition frequency.

The next factor is the charging and discharging speed of the circuit. In order to achieve a high repetition rate output, a lower capacitance is required. However, under the same conditions, the smaller the capacitance, the smaller the amplitude of the pulse. To balance these two factors, generally a capacitance value below 100 pF is used. When operating at higher frequencies, the maximum value of limiting resistance can only be several thousand ohms due to energy loss during charging circuits for high-frequency signals. Therefore, low-loss capacitors with high frequency should be selected for charging capacitors while large power non-inductive resistors should be chosen for limiting resistance. In addition, fast recovery diodes can replace limiting resistors to reduce energy loss and improve repetition rate. The speed of pulse generation circuit's charge–discharge process is limited by its charge resistor, storage capacitor and structural form of circuitry. It is best to choose anti-pulse solid ceramic resistors without inductance as limiting resistance while reducing system time constant through constant current charging method.

Finally, let's talk about power supply and circuit heat dissipation. The circuit working in a high-frequency and high-voltage mode will lead to an increase in the output power of the power supply. Therefore, a suitable high-voltage power supply should be selected to avoid insufficient output power of the power supply resulting in a decrease or no output voltage amplitude of the driving module. In this experiment, when operating at 1 Hz, only 700 V voltage is needed to obtain a pulse signal with an amplitude of 1.2 kV at the output end. However, as the frequency increases, the required input power of the circuit also increases while energy loss gradually becomes significant. Therefore, it is necessary to increase the power supply voltage to 900 V to maintain the same voltage amplitude. On another note, designing narrower pulse outputs can reduce load on the power supply and achieve higher output frequencies without changing its capacity for supplying electricity. Circuit heat dissipation is equally important because when working at high frequencies and voltages, overall circuitry consumes large amounts of energy which causes device

temperatures to rise rapidly; if there isn't good heat dissipation environment available, then component burnout may occur which could affect experimental safety.

## 5. Conclusions

In this paper, we design a new Marx structure and a high-repetition frequency UPG (ultrafast electric pulse generation) circuit. At the same time, the output is made more stable by matching optimization and module improvement, which solves the problems of high-repetition frequency ultrafast high-voltage pulse generation, ultra-broadband ultrafast pulse transmission and circuit stability and applicability, and can obtain high-repetition frequency ultrafast high-voltage electrical pulse with good quality at the load side. In the end 50 Ω load, a high-repetition frequency of 200 kHz, 800 ps leading edge, 5 ns FWHM and 1.2 kV amplitude can be obtained with less than 3% jitter.

Our analysis is of the output results, including its waveform and re-frequency. As the next step to improve the direction of work, the following are suggested as directions for improvement: optimization of the theoretical model to make it more refined; optimization of the peripheral circuit design of the re-frequency so that it can play the limit of the triode level as much as possible; optimization of the circuit matching by reducing the various distribution parameters, so as to improve the conduction speed; process direction of the exploration of higher performance switching devices such as FID, DSRD, etc.; reasonable PCB routing to minimize losses during transmission as much as possible; more attempts in stability.

This technology research can be used in streak camera scanning circuit to enable high re-frequency scanning, which can be used in the field of weak light detection to effectively capture the target information with weak luminous intensity and low quantum yield. It can also be used as a source of interference for electronic devices for electromagnetic interference technology, anti-interference stress testing of various electronic products, continuous improvement of the stability and accuracy of various devices working in strong electromagnetic environments and continuous updating of interference and anti-interference technology. The next step of this research is vertical for improvements on the repetition frequency, rise-time speed and pulse amplitude and horizontal for studying broadband bipolar pulses and chopper circuits for better performance.

**Author Contributions:** Conceptualization, H.X.; methodology, H.X.; Software, H.X.; Validation, H.X., P.F. and X.W.; Formal analysis, H.X.; Investigation, H.X.; Resources, H.X.; Data curation, H.X.; Writing—original draft preparation, H.X.; Writing—review and editing, H.X. and Y.Y.; Visualization, H.X. and S.W.; Supervision, B.L. and Y.G.; Project administration, B.L. and Y.G.; Funding acquisition, J.T., B.L. and Y.G. All authors have read and agreed to the published version of the manuscript.

**Funding:** This research was funded by National Natural Science Foundation of China (grant number No. 12075311 and 11805267), The Scientific Instrument Developing Project of the Chinese Academy of Sciences (grant number No. GJJSTD20220006), Strategic Priority Research Program of Chinese Academy of Sciences (grant number No. XDA25030900), Youth Innovation Promotion Association CAS (grant number No. 2021402).

**Acknowledgments:** Thank the Key Laboratory of Ultrafast Photoelectric Diagnostics Technology, Xi'an Institute of Optics and Precision Mechanics, Chinese Academy of Sciences and Center of Materials Science and Optoelectronics Engineering, UCAS for supporting this research.

**Conflicts of Interest:** The authors declare no conflict of interest.

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
