# Peer review of "Research on Triode Based High Re-Frequency Ultrafast Electrical Pulse Generation Technology"

_electronics, doi:10.3390/electronics12081950_

Round 1
Reviewer 1 Report
The authors show the new circuit for ultrafast electric pulse generation. In my opinion, the results are interesting and have scientific potential.
The paper needs a few minor upgrades:
1. Figures 1-4 add nothing to the paper. They should be erased.
2. The captions of the figures should include dots (".") after the numbers.
3. The style of the references should be adapted to the requirements of the journal e.g. it is necessary to add dots after the initials of the names.
4. The definition of the avalanche triode (line 131) is not insufficient
5. The form "is when" (e.g. line 138) is informal in my opinion. The sentences with "is when" should be transformed.
6. The line 134 should be ended with a literature reference.
7. Also the literature reference should appear after "Marx configuration" in line 147
8. The sentence from line 160 "In line..." should be split into shorter sentences.
9. The space should be added to the figures' references (e.g. Figure 5 instead of Figure5)
10. In my opinion the schemes in Figure 5 should be marked as "a)", and "b)", and they should be included in the caption.
11. The description of the experimental setup in paragraph 3.2 is in not typically grammatical form. In my opinion, this should be transformed, and it would be nice to see a picture of the experimental setup, and a scheme.
4. Technical oversights:
Line 102 - there is capital B just after ",";
Line 131, 143 - There should be a dot before the capital letter;
Line 381 - There should be no capital letter
Author Response
- Figures 1 to 4 have been deleted;
- Have added the dot after the number of the ‘Figure’;
- The reference format has been modified;
- Have expanded definition of avalanche triode;
- Have revised the sentences;
- Have modified;
- Have modified;
- Have revised to short sentences;
- Space has been added after the figure citation in the full text;
- Figure 5 has been divided into Figure 5(a) and Figure 5(b);
- 2 and related expressions have been revised, and experimental equipment and pictures have been added;
- Letter case and punctuation are corrected.
Reviewer 2 Report
Please see the attachment.

Author Response
- The figures that are not very relevant to this article have been deleted, and the clarity of the related pictures has been improved;
- Figure 5 has been divided into Figure 5(a) and Figure 5(b);
- Have added comment for trigger and output on the graph;
- The output waveform is not a simulation, it is a .csv file exported from the oscilloscope and generated by Origin;
- Experimental equipment and design process have been added to the paper;
- More technical terms have been replaced;
- Some complex phrases have been replaced with abbreviations;
- Some long sentences have been simplified to make them more understandable;
- Device vendor information has been added and Data Sheet has been added to the attachment;
- ‘Tube’ is due to the usual abbreviation habit, has been modified to transistor;
- Terminology has been added to the article.
Reviewer 3 Report
This article has serious flaws and research not conducted correctly. Some comments are listed as follows:
-Figures and equations which are not belong to authors should be cited.
-Quality of figures should be improved. The written text on the figures should be readable in printed version or 100% zoom. (Fig.9, Fig5, Fig.6)
- Only simulations are reported. Provide fabricated device and experimental results.
-Comparison table should be added.
-Provide schematic diagram of matching network and improvement module and add explanations about this part.
-Add conventional Marx structure and compare this structure with proposed design.
-The introduction section does not provide sufficiently and this section should be rewritten.
-The methods are not adequately described. The design flowchart should be added.
-The results are not presented clearly.
The abstract and conclusions should be rewritten and they should supported by the results.
Author Response
- Have added references to formulas and remove redundant images;
- Figures have been recreated;
- Experimental equipment and experimental results have been provided;
- Cross-reference tables have been added;
- Comparison chart has been provided;
- Have added comparison;
- The introduction has been rewritten;
- Have added flow chart;
- Have optimized statement expressions;
- Have optimized summary and conclusions.
Round 2
Reviewer 2 Report
The paper sounds more like a lab report!
I could not find any Optimization, analysis discovery or new insights.
Author Response
Thanks for your comments and suggestions.
I made the following changes to my article:
- I Have expanded circuit design optimization and PCB design section with additional specific optimization and reasons to the Circuit Design and Implementation (Chapter 3.1);
- I have added the analysis of waveform trailing to the Discussion (Chapter 4);
- I have optimized the conclusions and added an analytical summary of the experiments to the Conclusions (Chapter 5).
Reviewer 3 Report
The revised manuscript is improved, compared to the original version and quality of figure are improved, physical PCB is provided and etc.
Overall, most of my concerns have addressed by authors.
Author Response
Thanks for your comments and suggestions.
I further optimized the description of the circuit improvements, added new analysis, and optimized the summary.
Round 3
Reviewer 2 Report
Please see the attachment.

Author Response
Thanks very much for your time to review this manuscript. I really appreciate all your comments and suggestions. We have considered these comments carefully and tried our best to address every one of them.
Firstly, we have corrected the grammar errors and sentence structure according to your suggestions;
Secondly, we have rewritten a large amount of content in the Discussion section of Chapter 4 according to your suggestion, making it more profound and specific.
Round 4
Reviewer 2 Report
Please see the attachment.

Author Response
Thanks very much for your time to review this manuscript. I really appreciate all your comments and suggestions. We have considered these comments carefully and tried our best to address every one of them.
Firstly, we have corrected the grammar errors and sentence structure to make it more academically standardized according to your suggestions;
Then, we have deleted a large amount of meaningless content in the Discussion section of Chapter 4 as per your suggestion, and rewritten some parts to make it more research-oriented.